# Prevalence and correlates of microalbuminuria in high-risk persons with hypertension, diabetes, or HIV at a tertiary hospital in Zambia

Luyando Mutelo[1,2], Cornelius Simutanda[2], Lweendo Muchaili[1,2], Bislom C. Mweene[1], Situmbeko Liweleya[2,3], Sydney Mulamfu[2], Gift C. Chama [1], Lukundo Siame [1,2], Benson M. Hamooya[2,4], Sepiso K. Masenga [1,2,3]*

**1** Department of Pathology and Microbiology, Mulungushi University, School of Medicine and Health Sciences, Livingstone, Zambia, **2** Department of Cardiovascular Science and Metabolic diseases, Livingstone Center for Prevention and Translational Science, Livingstone, Zambia, **3** Department of Physiological Sciences, Mulungushi University, School of Medicine and Health Sciences, Livingstone, Zambia, **4** Department of Public Health, Mulungushi University, School of Medicine and Health Sciences, Livingstone, Zambia

* sepisomasenga@lcpts.org

## Abstract

### Background

Microalbuminuria is a critical marker for early kidney damage and a predictor for proteinuria, a defining feature of renal disease. Its detection in high-risk individuals is vital for reducing the risk of adverse renal outcomes, particularly among persons living with diabetes, hypertension, and HIV. The evaluation of microalbuminuria in high-risk individuals allows for the early detection of renal impairment and the assessment of cardiovascular risk profiles. Additionally, it informs targeted therapeutic interventions and provides critical prognostic information, thereby enhancing overall clinical outcomes. The goal of the study was to determine the prevalence and correlates of microalbuminuria in high risk patients.

### Methods

A cross-sectional study was conducted at Livingstone University Teaching Hospital (LUTH) from September 2023 to July 2024. We employed homogeneous purposive sampling to recruit 306 high-risk adults (≥18 years) with diabetes, hypertension, and/or HIV. Purposive sampling was employed to recruit patients attending routine medical clinic at LUTH. Sample size was calculated using the formula for a single proportion (expected prevalence: 30%, margin of error: 5%, 95% CI), yielding 323; we recruited 306 due to logistical constraints. Data were collected directly from participants as well as using hospital records and laboratory analysis of blood/urine samples. Descriptive and inferential statistics were analyzed using SPSS v22.

**Data availability statement:** The data underlying the results presented in the study are contained within the manuscript and/or Supporting information files S2.

**Funding:** This work was supported by the Fogarty International Center and National Institute of Diabetes and Digestive and Kidney Diseases of the National Institutes of Health grants R21TW012635 (SKM), the International Federation of Clinical Chemistry and Laboratory Medicine's Task Force on Outcome Studies in Laboratory Medicine (TF-OSLM) (SKM) and the American Heart Association Award Number 24IVPHA1297559 https://doi.org/10.58275/AHA.24IVPHA1297559.pc.gr.193866 (SKM). The content is solely the responsibility of the authors and does not represent the official views of the National Institutes of Health, IFCC and the American Heart Association. The funders had no role in the study design, data collection and analysis, decision to publish, or preparation of the manuscript.

**Competing interests:** The authors have declared that no competing interests exist.

## Results

A total of 306 high-risk individuals were screened, with a median age of 46 years (IQR: 31–58). Of these, 61.8% (n = 189) were females and 38.2% (n = 117) were males. The prevalence of microalbuminuria was 28.8% (95%CI 23.7–33.8), while its prevalence among individuals with diabetes, hypertension, and HIV was 45.4%, 33.3%, and 39.1%, respectively. Microalbuminuria was significantly associated with diabetes (p = 0.033), individuals living with HIV (p = 0.035), and high-sensitivity C-reactive protein (Hs-CRP) (p = 0.007) in univariate analysis. On multivariable analysis, microalbuminuria was associated with diabetes (AOR 2.90; 95% CI 1.02–8.19 p = 0.044), Living with HIV (AOR 1.96; 95%CI 1.02–3.78; p = 0.042) and Hs-CRP (AOR 1.16; 95%CI 1.02–1.30 p = 0.015).

## Conclusion

The findings underscore the importance of routine screening for microalbuminuria in high-risk populations. Early detection and intervention could prevent the progression of microalbuminuria to overt renal disease, particularly in individuals with diabetes, HIV, and systemic inflammation. Integrating microalbuminuria screening into chronic disease management programs is essential to mitigate renal complications in these vulnerable groups.

## Introduction

Microalbuminuria is urinary albumin excretion in the range of 30–300 mg per 24 hours and is considered as an abnormal albumin excretion rate which is associated with endothelial dysfunction in the kidneys and has a high risk for target organ damage resulting in stroke, retinopathy and adverse cardiovascular events [1]. Microalbuminuria increases the risk for cardiovascular morbidity and mortality especially in high-risk populations such as those living with diabetes and hypertension [2]. HIV infection and use of antiretroviral therapy (ART) often impacts multiple organs, with the kidneys being a common target, making renal disease a recognized complication. Microalbuminuria serves as an early indicator of kidney damage, particularly in ART-naïve people living with HIV (PLWH) [3]. Early detection is crucial for preventing progression to chronic kidney disease (CKD) in this population [3]. One study from Zambia reported the prevalence of CKD due to hypertension and diabetes to be 18.7% and 22.7%, respectively [4]. The capacity of microalbuminuria to predict an increased cardiovascular and renal risk is well established in diabetic patients [5]. Additionally, Microalbuminuria is frequent in PLWH as a predictor of renal impairment and CV risk [6].

The prevalence of microalbuminuria varies across high-risk populations, with rates estimated between 6% and 47% depending on specific patient cohorts and comorbidities [7,8]. Elevated microalbuminuria levels in such high-risk cohorts highlight the need for timely intervention to prevent progression to acute kidney injury and other renal complications. Despite its well-recognized prognostic value, most health

care institutions lack the diagnostic capacity to screen for microalbuminuria and there remains a need for comprehensive understanding of the prevalence and correlates of microalbuminuria specifically among patients at high risk for kidney injury.

This study aimed to determine the prevalence and associated factors of microalbuminuria in high-risk individuals living with HIV, diabetes and hypertension. By identifying the correlates of microalbuminuria within these populations, this study aimed to contribute valuable insights for early detection and intervention strategies, thereby improving outcomes and potentially reducing the incidence of kidney disease in high-risk groups.

## Methods

### Study design

This was a single-center cross-sectional study conducted at Livingstone University Teaching Hospital (LUTH). We screened 306 high-risk paticipants between September 1, 2023 and 30th July 2024.

### Eligibility and recruitment

We recruited adults aged 18 years and above with known high-risk conditions such as PLWH, hypertension and diabetes who were attenting routine clinic visits at the Medical clinic at LUTH. Paticipants without high-risk conditions, participants with prior diagnosis of kidney disease and those with missing age were excluded. Homogenous purposive sampling was employed. Participants were intentionally selected to create a homogeneous group sharing key characteristics (presence of diabetes, hypertension, or HIV). This reduced variability from irrelevant factors (e.g., non-high-risk individuals) and focused on the study's high-risk profile, enhancing internal validity.

### Sample size calculation

Sample size was determined using sample size formula for proportions where: $Z = 1.96$ (95% CI), $p = 0.30$ (maximum expected microalbuminuria prevalence regardless of disease using previous hospital records from medical clinic department), $d = 0.05$ (margin of error). This yielded 323 participants. We enrolled 306 due to time/resource constraints, achieving a 5.1% margin of error for the observed prevalence (28.8%).

### Sampling frame

The sampling frame included all eligible adults attending routine medical clinics at LUTH during September 2023–July 2024. Recruitment ceased at 306 after exhausting the study period.

### Variables

The outcome variable was microalbuminuria which was defined according to the American diabetes association using 24-hour collection technique as urine albumin concentration of 30–299 mg per 24 hours [9].

Independent variables included sociodemographic and lifestyle (age, sex, employment status, takes alcohol), clinical (Hypertension, diabetes, PLWH), Anthropometrics (Body mass index, waist circumference), laboratory tests (fasting blood sugar, plasma sodium, plasma urea, plasma creatinine, hs-CRP, Urine Micro-albumin). Diabetes was defined as a physician-diagnosed condition or current use of glucose-lowering medication. Hypertension was considered present if there was a physician diagnosis or use of antihypertensive medication. HIV status was determined based on confirmed HIV-positive diagnosis or use of antiretroviral therapy for HIV.

### Data source

We collected data from participants and hospital patient records were used to collect additional patient information. In addition, blood and urine sample collection was done for laboratory analysis.

## Data analysis

We used SPSS version 22 for data analysis. Descriptive statistics was used to understand the distribution of variables among the study participants. Mann-Whitney or t-test was used to compare medians of two continuous variables depending on the normality of data. The Chi-square test was employed to assess the relationship between two categorical variables. We used univariable and multivariable logistic regression to investigate factors associated with microalbuminuria. We described the data using frequencies and proportions.

## Ethics

Ethical approval was obtained from Mulungushi University School of Medicine and Health Sciences Research Ethics Committee (MUSoMHSREC) (Assurance No. FWA0002888 IRB00012281 of IORG0010344 on the 11th of August 2023 and the National Health Research Authority (NHRA) on the 23rd of August, 2023. The LUTH's administration permitted to conduct research. This study is part of the ICLATA project (Impact of Implementing a Complete Clinical Laboratory Test Profile on Clinical Outcomes of Acute Kidney Injury: The ICLATA Study).

Written consent was obtained, and all data were de-identified to ensure that no information collected could be linked to any specific participant. To strengthen the reporting for this observational study, we used the Strengthening the Reporting of Observational Studies in Epidemiology (STROBE) for reporting (S1 File). All data underlying the study are available (S2 File).

## Results

### Sociodemographic and clinical factors associations between those with and without Microalbuminuria

The study involved 306 high-risk paticipants, of which 61.8% (n = 189) were female and 38.2% (n = 117) were male. The prevalence of microalbuminuria was 28.8% (23.7%, 33.8% 95%CI). Condition-specific prevalences were: diabetes 45.2% (95% CI: 27.6%–62.7%), hypertension 33.3% (95% CI: 24.0%–42.6%), and HIV 39.1% (95% CI: 29.2%–49.1%). The median age at diagnosis was significantly higher for individuals with microalbuminuria (48 years; IQR: 37–60) compared to those without microalbuminuria (45 years; IQR: 30–55; P = 0.031). The majority with microalbuminuria were employed 57.9% (n = 11). Alcohol intake accounted for the majority, with microalbuminuria 43.2% (n = 19). Overall, microalbuminuria was more prevalent in females 29.3% (n = 55) not males (28.2%, n = 33) but was not associated with sex, while age showed a significant association (Table 1).

Alcohol consumption, Body Mass Index (BMI), and waist circumference were comparable between individuals with and without microalbuminuria, showing no significant differences. However, individuals with microalbuminuria exhibited higher systolic blood pressure (SBP) (median 137 mmHg vs. 128 mmHg), while diastolic blood pressure (DBP) did not differ significantly between the groups. The prevalence of hypertension was also similar between those with and without microalbuminuria.

A higher proportion of individuals with microalbuminuria were living with diabetes. Fasting blood sugar, plasma sodium, plasma urea, and plasma creatinine levels were not significantly different between those with and without microalbuminuria.

Living with HIV was associated with microalbuminuria. High-sensitivity C-reactive protein (Hs-CRP) levels were significantly elevated in individuals with microalbuminuria, suggesting heightened systemic subclinical inflammation in this group.

### Sociodemographic and clinical factors associated with microalbuminuria

On univariable and multivariable logistic regression models, diabetes, HIV status, and elevated hs-CRP levels were significantly associated with microalbuminuria, **Table 2**. Those with Diabetes were 2.9 times more likely to have microalbuminuria compared to those without diabetes (**AOR:** 2.90; 95% **CI:** 1.02–8.19; **P = 0.044**), while individuals living with HIV were 1.96 times more likely to have microalbuminuria compared to individuals without HIV (**AOR:** 1.96; 95% **CI:**

**Table 1. Sociodemographic and clinical factors between those with and without Microalbuminuria.**

| Variables | Frequency (%) or Median (IQR) | MicroalbuminuriaYES, n=88 (28.8%) | Microalbuminuria, NO, n=218 (71.2%) | P-Value |
|---|---|---|---|---|
| **Age,** *years* | 46 (31-58) | 48 (37-60) | 45 (30-55) | **0.031** |
| **Sex,** *n=305* | | | | |
| *Male* | 117 (38.2) | 33 (28.2) | 84 (71.8) | 0.844 |
| *Female* | 189 (61.8) | 55 (29.1) | 134 (70.9) | |
| **Marital status,** *n=57* | | | | |
| *Married* | 34 (59.6) | 14 (41.2) | 20 (58.8) | 0.862 |
| *Unmarried* | 23 (40.4) | 10 (43.5) | 13 (5.5) | |
| **Employment status,** *n=59* | | | | |
| *Employed* | 19 (32.2) | 11 (57.9) | 8 (42.1) | 0.063 |
| *Unemployed* | 40 (67.8) | 13 (32.5) | 27 (67.5) | |
| **Takes alcohol,** *n=59* | | | | |
| *No* | 44 (74.6) | 19 (43.2) | 25 (56.8) | 0.502 |
| *Yes* | 15 (25.4) | 5 (33.3) | 10 (66.7) | |
| **Body mass index, kg/m²** | 25.9 (22.2-29.78) | 24.7 (20.9-29.7) | 26.2 (22.9-29.8) | 0.152 |
| **Waist circumference,** *cm; n=289* | 81 (50-96 | 80.7 (70-99) | 82 (40-95) | 0.294 |
| **SBP** | 129 (116-146) | 137 (117-152) | 128 (115-145) | **0.024** |
| **DBP** | 82 (73-93) | 81.5 (73-95.5) | 82 (73-93) | 0.720 |
| **Hypertension, n=304** | | | | |
| *No* | 205 (67.4) | 55 (26.8) | 150 (73.2) | 0.241 |
| *Yes* | 99 (32.6) | 33 (33.3) | 66 (66.7) | |
| **Diabetes** | | | | |
| *No* | 275 (89.9) | 74 (26.9) | 201 (73.1) | **0.033** |
| *Yes* | 31 (10.1) | 14 (45.2) | 17 (54.8) | |
| **Duration of diabetes** | 10 (1-13) | 12 (6-15) | 7 (1-10) | 0.460 |
| **Fasting blood sugar,** *n=136* | 4.9 (4.4-5.5) | 4.8 (4.5-6) | 4.9 (4.6-5.5) | 0.763 |
| **Living with HIV** | | | | |
| *No* | 93 (50.3) | 23 (24.7) | 70 (75.3) | **0.035** |
| *Yes* | 92 (49.7) | 36 (39.1) | 56 (60.9) | |
| **Plasma sodium,** *n=245* | 143 (138-145) | 144 (139-146) | 143 (138-145) | 0.114 |
| **Plasma Urea,** *n=117* | 3.28 (2.56-4.27) | 3.75 (2.65-5.51) | 3.25 (2.53-3.85) | 0.074 |
| **Plasma creatinine, μmol/l,** *n=239* | 63.79 (53.93-76.8) | 64.3 (55.6-77.1) | 63.1 (52.5-76.78) | 0.449 |
| **eGFR,** mL/min/1.73m² n=239 | 108 (94, 123) | 103 (93, 117) | 109 (95, 124) | 0.056 |
| **Hs-CRP** | 3.60 (0.83-10) | 7.17 (2.29-10.0) | 2.54 (0.48-10) | **0.007** |

**SBP**; Systolic Blood Pressure, **DBP**; Diastolic Blood Pressure, **hs-CRP**; High Sensitivity C-Reactive Protein

1.02–3.78; *P*=0.042). Elevated hs-CRP levels were associated with a 16% increase in odds for each unit increase of having microalbuminuria (**AOR:** 1.16; 95% **CI:** 1.02–1.30; *P*=0.015). Although age and systolic blood pressure (SBP) were significantly associated with microalbminuria in unadjusted models, the relationship was abrogated after adjustment (*P*>0.05). These findings emphasize the interplay between metabolic, infectious, and inflammatory factors in the development of microalbuminuria.

**Table 2. Sociodemographic and clinical factors associated with microalbuminuria in logistic regression.**

| Variables | OR (95%CI) | P-Value | AOR (95%CI) | P-Value |
|---|---|---|---|---|
| **Age,** *years* | 1.01 (1.00-1.03) | **0.022** | 0.99 (0.95-1.02) | 0.616 |
| **SBP** | 1.01 (1.00-1.02) | **0.013** | 1.01 (0.99-1.04) | 0.083 |
| **Diabetes** | | | | |
| *No* | 1 | | 1 | |
| *Yes* | 2.23 (1.05-4.76) | **0.036** | 2.90 (1.02-8.19) | **0.044** |
| **Living with HIV** | | | | |
| *No* | 1 | | 1 | |
| *Yes* | 1.95 (1.04-3.67) | **0.036** | 1.96 (1.02-3.78) | **0.042** |
| **Hs-CRP** | 1.15 (1.03-1.28) | **0.012** | 1.16 (1.02-1.30) | **0.015** |

**SBP**; Systolic Blood Pressure, **hs-CRP**; High Sensitivity C-Reactive Protein

## Discussion

The study aimed to determine the prevalence and factors associated with microalbuminuria. The combined prevalence of microalbumnuria was 28.8% while the prevalence among individuals with diabetes, hypertension, and HIV was 45.4%, 33.3%, and 39.1%, respectively.

The prevalence of microalbuminuria among diabetic individuals in this study was 45.4%, which is marginally higher than the 37.11% reported in a systematic review and meta-analysis conducted in Africa among individuals with diabetes [10]. Similarly, another study [11] found the prevalence of microalbuminuria among diabetic participants was 29.72%. Our study revealed that individuals with diabetes were 2.9 times more likely to have microalbuminuria compared to those without diabetes, emphasizing the strong link between diabetes and kidney damage. Similarly, *Basi et al* [12] revealed that participants with diabetes had a relative ratio (RR) of 1.97 and those without diabetes had an RR of 1.61 indicating that albuminuria is a presage of renal risk in patients with diabetes.

Research indicates that microalbuminuria is also frequently observed in patients with established essential hypertension and serves as a high risk predictor of cardiovascular events [13]. The positive relationship between SBP and microabumnuria in our study has been reported previously in a cross-sectional study by *Knight et al* [14] as well as in a meta analysis study [15] that revealed that persons with hypertension were at risk for developing microalbuminuria which was associated with an increased risk of all-cause mortality. These studies highlight the need for monitoring SBP, including nocturnal measurements in the early identification of individuals at risk for developing kidney disease or kidney injury. The prevalence of microalbuminuria among persons with hypertension in this study was 33.3%. This prevalence is higher than the study conducted in Zambia by *Rasmussen et al* [16] where they found the prevalence of microalbuminuria in hypertensive individuals was 19.6%. Another study by *satoru et al* [17] reported a prevalence of 31.6%. The highlight of these findings is the notable prevalence of microalbuminuria among persons with hypertension which underscores the need for routine screening and early interventions to prevent the progression of renal and cardiovascular complications.

Microalbuminuria is an independent risk factor for cardiovascular and kidney disease and a predictor of end organ damage, both in the general population and in persons with HIV [18]. We found the prevalence of microalbuminuria among PLWH was 39.1%. This prevalence was higher compared to other studies, with prevalence rates of 20.7%, 24%, and 28.8% observed in Botswana, South Africa, and Tanzania, respectively [19,20]. Our study also revealed that PLWH were 1.96 times likely to have microalbuminuria. The odds were lower than what *Szczech et al* [21] reported in their HIV cohort (fivefold increased odds). The study highlights global variability in the prevalence of microalbuminuria among PLWH, reflecting differences in healthcare and population factors. Nonetheless, these findings underscore the need for early detection, tailored interventions, and effective public health strategies to address kidney disease risk in PLWH.

Furthermore, Hs-CRP, a marker of systemic inflammation, showed that with each unit increase, the odds of having microalbuminuria increased by about 15–16%, this finding suggests that inflammation plays a crucial role in the development of microalbuminuria. A similar findings was reported in the National Health and Nutrition Examination Surveys (NHANES) [22]. Research done by *Yang et al* [23] indicated that the progression of microalbuminuria was more pronounced in individuals with elevated baseline hs-CRP which also aligns with another study by *Marcovecchio et al* [24]. This demonstrates that higher baseline hs-CRP levels, along with diabetes, are predictive of microalbuminuria progression.

## Strengths and limitations

The use of urine spot tests (micro-albumin) highlighted its effectiveness in early detection of kidney injury in high-risk patients. The study also addressed emerging risk factors, this study provides a robust foundation for advancing knowledge, informing policy, and improving clinical outcomes related to microalbuminuria. Our study is distinguished by unique population characteristics. Our study contributes novel insights by simultaneously evaluating three high-risk groups: diabetes, hypertension, and HIV, within a single Zambian cohort from a tertiary teaching hospital serving urban/peri-urban communities. This population is underrepresented in microalbuminuria research despite bearing a triple burden of communicable (HIV) and non-communicable diseases (diabetes/hypertension) amid resource constraints. The homogeneous sampling strategy specifically targeted these comorbidities, reducing confounding from non-risk factors and providing focused data on their synergistic effects. Additionally, Zambia's distinct genetic, socioeconomic, and healthcare access profiles may influence microalbuminuria risk differently than populations in prior studies, underscoring the urgency of context-specific screening protocols in similar sub-Saharan African settings.

Our study is not without limitations. Our study was cross sectional and could not determine causality. Further, a large sample size addressing the three key populations (PLWH, diabetes and hypertension) would be required to validate our findings. Our study was not powered enough to accurately perform sub-analysis comparing PLWH, diabetes and hypertension. As a single-center study at a tertiary hospital, findings may not represent national prevalence. Caution is warranted when generalizing beyond similar urban/peri-urban settings. Future studies should incorporate HbA1c and urine protein-to-albumin ratios to refine risk stratification.

## Conclusions

Microalbuminuria testing plays a critical role in the early detection of kidney damage. We have found a strong association between diabetes, HIV, and elevated microalbuminuria levels, indicating an increased risk of kidney injury in these populations. Furthermore, systemic inflammation, as reflected by elevated Hs-CRP levels, is linked to a heightened risk of developing microalbuminuria. These findings underscore the importance of early detection and regular monitoring to prevent the progression of renal disease.

## Supporting information

**S1 File. STROBE Statement—Checklist of items that should be included in reports of *cross-sectional studies*.**
(DOCX)

**S2 File. Microalbuminuria DATA.**
(XLSX)

## Acknowledgments

We thank the HAND Research Group at Mulungushi University and the Livingstone Center for Prevention and Translational Science for all the support rendered during the study, as well as the management of LUTH for permitting us to conduct this study.

## Author contributions

**Conceptualization:** Luyando Mutelo, Cornelius Simutanda, Sepiso K. Masenga.

**Data curation:** Luyando Mutelo, Cornelius Simutanda, Sepiso K. Masenga.

**Formal analysis:** Luyando Mutelo, Sepiso K. Masenga.

**Funding acquisition:** Sepiso K. Masenga.

**Investigation:** Luyando Mutelo, Cornelius Simutanda, Sepiso K. Masenga.

**Methodology:** Luyando Mutelo, Sepiso K. Masenga.

**Project administration:** Cornelius Simutanda, Sepiso K. Masenga.

**Resources:** Sepiso K. Masenga.

**Software:** Sepiso K. Masenga.

**Supervision:** Lweendo Muchaili, Bislom C. Mweene, Situmbeko Liweleya, Sydney Mulamfu, Benson M. Hamooya, Lukundo Siame, Sepiso K. Masenga.

**Validation:** Cornelius Simutanda, Lweendo Muchaili, Bislom C. Mweene, Situmbeko Liweleya, Sydney Mulamfu, Gift C. Chama, Benson M. Hamooya, Lukundo Siame, Sepiso K. Masenga.

**Visualization:** Benson M. Hamooya, Lukundo Siame, Sepiso K. Masenga.

**Writing – original draft:** Luyando Mutelo, Cornelius Simutanda, Lweendo Muchaili, Bislom C. Mweene, Situmbeko Liweleya, Sydney Mulamfu, Gift C. Chama, Benson M. Hamooya, Lukundo Siame, Sepiso K. Masenga.

**Writing – review & editing:** Luyando Mutelo, Cornelius Simutanda, Lweendo Muchaili, Bislom C. Mweene, Situmbeko Liweleya, Sydney Mulamfu, Gift C. Chama, Benson M. Hamooya, Lukundo Siame, Sepiso K. Masenga.

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
