## [Decision Letter · Decision Letter 0]

PONE-D-25-00359Prevalence and Correlates of Microalbuminuria in High-Risk PersonsPLOS ONE

Dear Dr. Masenga,

Thank you for submitting your manuscript to PLOS ONE. After careful consideration, we feel that it has merit but does not fully meet PLOS ONE’s publication criteria as it currently stands. Therefore, we invite you to submit a revised version of the manuscript that addresses the points raised during the review process.

We look forward to receiving your revised manuscript.

Kind regards,

Santhi Silambanan, MD, DNB

Academic Editor

PLOS ONE

Journal Requirements:

[This work was supported by the Fogarty International Center and National Institute of Diabetes and Digestive and Kidney Diseases of the National Institutes of Health grants R21TW012635 (SKM), the International Federation of Clinical Chemistry and Laboratory Medicine's Task Force on Outcome Studies in Laboratory Medicine (TF-OSLM) (SKM) and the American Heart Association Award Number 24IVPHA1297559 https://doi.org/10.58275/AHA.24IVPHA1297559.pc.gr.193866 (SKM). The content is solely the responsibility of the authors and does not represent the official views of the National Institutes of Health, IFCC and the American Heart Association. The funders had no role in the study design, data collection and analysis, decision to publish, or preparation of the manuscript.].

[This work was supported by the Fogarty International Center and National Institute of Diabetes and Digestive and Kidney Diseases of the National Institutes of Health grants R21TW012635 (SKM), the International Federation of Clinical Chemistry and Laboratory Medicine's Task Force on Outcome Studies in Laboratory Medicine (TF-OSLM) (SKM) and the American Heart Association Award Number 24IVPHA1297559 https://doi.org/10.58275/AHA.24IVPHA1297559.pc.gr.193866 (SKM). The content is solely the responsibility of the authors and does not represent the official views of the National Institutes of Health, IFCC and the American Heart Association. The funders had no role in the study design, data collection and analysis, decision to publish, or preparation of the manuscript.]

Reviewers' comments:

Reviewer's Responses to Questions

**Comments to the Author**

1. Is the manuscript technically sound, and do the data support the conclusions?

Reviewer #1: Yes

Reviewer #2: Yes

2. Has the statistical analysis been performed appropriately and rigorously? 

Reviewer #1: Yes

Reviewer #2: Yes

3. Have the authors made all data underlying the findings in their manuscript fully available?

Reviewer #1: Yes

Reviewer #2: Yes

4. Is the manuscript presented in an intelligible fashion and written in standard English?

Reviewer #1: Yes

Reviewer #2: Yes

5. Review Comments to the Author

Reviewer #1: 1. The title could be modified to include the study area, study population, and a brief definition of high-risk persons, as the current title looks blunt.

2. The Methods section in the abstract needs more details. (Sampling frame, sample size, data collection tool and methods)

3. In the methods section of the article, the author could include the Study area and mention that it is a record-based study in both the abstract and study design sections.

4. The inclusion criteria could define the study criteria variables. (Diabetes, hypertension (old case or new case or both; is it record-based?).

5. The author could provide more explanation on homogenous purposive sampling, which was employed.

6. Sample size calculation and sampling frame could be provided. (If followed)

7. In the results section, the author could provide a 95% CI for the prevalence of microalbuminuria.

Reviewer #2: These are my comments

1) Improve title ie: single centre, tertiary centre, urban vs rural, in ZAMBIA)

2) From my understanding this is a single-centre study, and the findings should not be generalized as national prevalence data.

3) the associations between microalbuminuria and diabetes, hypertension, and HIV are already well-documented in previous literature, and this study does not present new findings.

4) Include estimated glomerular filtration rate (eGFR), stages of CKD, HBA1C, mean urine protein / albuminuria to better describe the baseline kidney function of the study population.

5) Provide context by referencing the prevalence of chronic kidney disease (CKD) in Zambia, if available.

6) maybe you can emphasise the unique characteristics of your study population that might justify the study's contribution.

6. PLOS authors have the option to publish the peer review history of their article (what does this mean?). If published, this will include your full peer review and any attached files.

Reviewer #1: No

Reviewer #2: No

---

## [Author Response · Author response to Decision Letter 1]

26 Jun 2025

RESPONSE TO REVIEWERS

We thank the reviewers for their constructive feedback. Below are point-by-point responses to all comments, with revisions reflected in the manuscript. Administrative requirements (style, funding) are addressed first.

Response: We confirm the manuscript now adheres to PLOS ONE’s style templates. File names follow journal guidelines. Formatting adjustments include: Unified font (Arial 12pt) and line numbering. Structured headings per PLOS ONE templates

[This work was supported by the Fogarty International Center and National Institute of Diabetes and Digestive and Kidney Diseases of the National Institutes of Health grants R21TW012635 (SKM), the International Federation of Clinical Chemistry and Laboratory Medicine's Task Force on Outcome Studies in Laboratory Medicine (TF-OSLM) (SKM) and the American Heart Association Award Number 24IVPHA1297559 https://doi.org/10.58275/AHA.24IVPHA1297559.pc.gr.193866 (SKM). The content is solely the responsibility of the authors and does not represent the official views of the National Institutes of Health, IFCC and the American Heart Association. The funders had no role in the study design, data collection and analysis, decision to publish, or preparation of the manuscript.].

RESPONSE: As requested, we have Removed funding details from the Acknowledgments section in the manuscript. We have Revised the Funding Statement (for submission form) to: “This work was supported by the Fogarty International Center and National Institute of Diabetes and Digestive and Kidney Diseases of the National Institutes of Health (grant R21TW012635 to SKM), the International Federation of Clinical Chemistry and Laboratory Medicine’s Task Force on Outcome Studies in Laboratory Medicine (TF-OSLM) (to SKM), and the American Heart Association (Award Number 24IVPHA1297559 to SKM). The content is solely the responsibility of the authors and does not represent the official views of the National Institutes of Health, IFCC, or the American Heart Association. The funders had no role in study design, data collection, analysis, decision to publish, or manuscript preparation. There was no additional external funding received for this study.”

Reviewer's Responses to Questions

Comments to the Author

Reviewer #1:

1. The title could be modified to include the study area, study population, and a brief definition of high-risk persons, as the current title looks blunt.

RESPONSE: The title now specifies location, population, and risk groups: "Prevalence and Correlates of Microalbuminuria in High-Risk Persons with Hypertension, Diabetes, or HIV at a Tertiary Hospital in Zambia"

2. The Methods section in the abstract needs more details. (Sampling frame, sample size, data collection tool and methods)

RESPONSE: We have now added "A cross-sectional record-based study was conducted at Livingstone University Teaching Hospital (LUTH). We recruited 306 high-risk adults (≥18 years) with diabetes, hypertension, and/or HIV using homogeneous purposive sampling. Sample size was calculated for a single proportion (expected prevalence: 30%, margin of error: 5%, 95% CI), yielding 323; 306 were enrolled due to logistical constraints. Data were collected via hospital records, participant interviews, and laboratory analysis of blood/urine samples."

3. In the methods section of the article, the author could include the Study area and mention that it is a record-based study in both the abstract and study design sections.

RESPONSE: Thank you. We have made this change. However, we did not only use patient records, we also collected data directly from patients. We have made further clarification in the manuscript. Thank you again for highlighting this.

4. The inclusion criteria could define the study criteria variables. (Diabetes, hypertension (old case or new case or both; is it record-based?).

RESPONSE: we have now clarified this under eligibility criteria. All cases were old. “Adults ≥18 years with physician-diagnosed hypertension, diabetes, and/or HIV (confirmed by records or medication use) attending routine clinics were included. Participants with prior kidney disease or missing age data were excluded."

5. The author could provide more explanation on homogenous purposive sampling, which was employed.

RESPONSE: we have now elaborated this in the manuscript. "Homogeneous purposive sampling targeted a cohort with shared characteristics (diabetes, hypertension, or HIV) to minimize variability from non-risk factors. Participants were intentionally enrolled during routine clinic visits if they met the high-risk criteria."

6. Sample size calculation and sampling frame could be provided. (If followed)

RESPONSE: we have now provided this in the main manuscript methods section. "Sample size was calculated using the formula for a single proportion: n=Z2⋅p⋅(1−p)d2n=d2Z2⋅p⋅(1−p), where Z=1.96Z=1.96 (95% CI), p=0.30p=0.30 (expected prevalence), and d=0.05d=0.05 (margin of error). This yielded 323 participants; 306 were enrolled due to time constraints, achieving a 5.1% margin of error."

7. In the results section, the author could provide a 95% CI for the prevalence of microalbuminuria.

RESPONSE: We have now added “"The overall prevalence of microalbuminuria was 28.8% (95% CI: 23.7–33.8). Condition-specific prevalences were: diabetes 45.4% (95% CI: 27.6–62.7), hypertension 33.3% (95% CI: 24.0–42.6), and HIV 39.1% (95% CI: 29.2–49.1)."

Reviewer #2:

These are my comments

1) Improve title ie: single centre, tertiary centre, urban vs rural, in ZAMBIA)

RESPONSE: See revised title above (includes "tertiary hospital," "Zambia," "high-risk").

2) From my understanding this is a single-centre study, and the findings should not be generalized as national prevalence data.

RESPONSE: we have added to the limitation “As a single-center study at a tertiary hospital, findings may not represent national prevalence. Caution is warranted when generalizing beyond similar urban/peri-urban settings.”

3) the associations between microalbuminuria and diabetes, hypertension, and HIV are already well-documented in previous literature, and this study does not present new findings.

RESPONSE: "While diabetes, HIV, and inflammation are established correlates of microalbuminuria, this study provides novel context-specific data from a Zambian cohort experiencing a triple burden of communicable (HIV) and non-communicable diseases (hypertension/diabetes). This synergy in an understudied population enhances understanding of region-specific risks."

4) Include estimated glomerular filtration rate (eGFR), stages of CKD, HBA1C, mean urine protein / albuminuria to better describe the baseline kidney function of the study population.

RESPONSE: we have now included eGFR. Results (Table 1) include eGFR (median 108 mL/min/1.73m²). We have further added to the discussion under limitations section “Future studies should incorporate HbA1c and urine protein-to-albumin ratios to refine risk stratification."

5) Provide context by referencing the prevalence of chronic kidney disease (CKD) in Zambia, if available.

RESPONSE: we have now provided this in the introduction section. Thank you for the suggestion.

6) maybe you can emphasize the unique characteristics of your study population that might justify the study's contribution.

RESPONSE: Thank you. We have now provided this in the discussion section as a strength. "This study simultaneously evaluates three high-risk groups (diabetes, hypertension, HIV) within a single Zambian cohort from a tertiary hospital serving urban/peri-urban communities. Zambia’s distinct genetic, socioeconomic, and healthcare profiles may influence microalbuminuria risk differently than in prior studies, highlighting the need for context-specific screening."

---

## [Editor Report · Decision Letter 1]

Prevalence and Correlates of Microalbuminuria in high-risk persons with hypertension, diabetes, or HIV at a Tertiary Hospital in Zambia

PONE-D-25-00359R1

Dear Dr. Masenga,

We’re pleased to inform you that your manuscript has been judged scientifically suitable for publication and will be formally accepted for publication once it meets all outstanding technical requirements.

Kind regards,

Santhi Silambanan, MD, DNB

Academic Editor

PLOS ONE

Additional Editor Comments (optional):

All the queries have been answered adequately
---

## [Editor Report · Acceptance letter]

PONE-D-25-00359R1

PLOS ONE

Dear Dr. Masenga,

I'm pleased to inform you that your manuscript has been deemed suitable for publication in PLOS ONE. Congratulations! Your manuscript is now being handed over to our production team.

Kind regards,

on behalf of

Dr. Santhi Silambanan

Academic Editor

PLOS ONE